

# Ants contribute to pollination but not to reproduction in a rare calcareous grassland forb

Michael Rostás[1,2], Felix Bollmann[2], David Saville[3] and Michael Riedel[2]

[1] Bio-Protection Research Centre, Lincoln University, Lincoln, Canterbury, New Zealand
[2] Julius-von-Sachs-Institute for Biosciences, Department of Botany II, University of Würzburg, Würzburg, Bavaria, Germany
[3] Saville Statistical Consulting Ltd, Lincoln, Canterbury, New Zealand

## ABSTRACT

The number of plants pollinated by ants is surprisingly low given the abundance of ants and the fact that they are common visitors of angiosperms. Generally ants are considered as nectar robbers that do not provide pollination service. We studied the pollination system of the endangered dry grassland forb *Euphorbia seguieriana* and found two ant species to be the most frequent visitors of its flowers. Workers of *Formica cunicularia* carried five times more pollen than smaller *Tapinoma erraticum* individuals, but significantly more viable pollen was recovered from the latter. Overall, the viability of pollen on ant cuticles was significantly lower ($p < 0.001$)—presumably an antibiotic effect of the metapleural gland secretion. A marking experiment suggested that ants were unlikely to facilitate outcrossing as workers repeatedly returned to the same individual plant. In open pollinated plants and when access was given exclusively to flying insects, fruit set was nearly 100%. In plants visited by ants only, roughly one third of flowers set fruit, and almost none set fruit when all insects were excluded. The germination rate of seeds from flowers pollinated by flying insects was $31 \pm 7\%$ in contrast to $1 \pm 1\%$ resulting from ant pollination. We conclude that inbreeding depression may be responsible for the very low germination rate in ant pollinated flowers and that ants, although the most frequent visitors, play a negligible or even deleterious role in the reproduction of *E. seguieriana*. Our study reiterates the need to investigate plant fitness effects beyond seed set in order to confirm ant-plant mutualisms.

## INTRODUCTION

Ants are frequent visitors of angiosperms, and although they may be involved in mutualistic interactions that benefit the plant such as seed dispersal or the removal of herbivorous insects, ants have traditionally been regarded as poor pollinators (*Beattie, 2006*; *Rico-Gray & Oliveira, 2007*; *Rostás & Tautz, 2011*). Accordingly, a relatively small number of ca. 40 supported cases of ant pollination have been documented so far, mostly within the last two decades (*De Vega & Gómez, 2014*).

Corresponding author
Michael Rostás,
michael.rostas@lincoln.ac.nz

Several traits associated with ants are held responsible for their insignificant or even detrimental role in plant reproduction. The main arguments against ants as vectors as facilitators of pollination in general are their foraging patterns and the presence of metapleural gland secretion on their integument. This secretion protects the ants from infection by microorganisms but may inadvertently have negative effects on the viability of pollen, except in some plants where ants are the primary pollinators (*Beattie et al., 1984*; *Beattie et al., 1986*; *De Vega et al., 2009*). *Dutton & Frederickson (2012)* compared several ant and plant species from temperate and tropical habitats and found reduced pollen germination in all cases with more pronounced effects in tropical ants, where microbes are expected to impose stronger selection for antibiotic defences.

Ant workers are wingless and small and it has been suggested that in many plants they do not carry pollen far enough for outcrossing and continuously return to the same individual plant for nectar. This *Ortstreue* (*Hölldobler & Wilson, 1990*) results predominantly in within-plant pollen movement and hence in geitonogamy. Not surprisingly, the large majority of ant pollinated plants listed in the latest review by *De Vega & Gómez (2014)* are considered to be self-compatible and would therefore be less affected by geitonogamy than obligate outcrossing species. Self-pollination can lead to the abortion of seeds in outcrossing plants but even in self-compatible species, inbred seeds may have lower germination rates and higher seedling mortality (*Baskin & Baskin, 2015*; *Gómez, 2000*). However, this aspect has been overlooked in almost all ant pollination studies so far and only *Gómez (2000)* has tested offspring viability (but see *Blancafort & Gómez, 2005* and *De Vega et al., 2009* for other measures of seed viability).

Ant pollinated species are often characterised by small, open flowers and by inhabiting dry and/or alpine habitats in which ants are generally abundant (*De Vega et al., 2009*; *Gómez et al., 1996*; *Hickman, 1974*). Indeed, this quantitative aspect of ant pollination has been identified as a crucial component, where numerous transfers of (often inviable) pollen grains to stigmata by many individuals can compensate for the vector's inefficiency (*Gómez & Zamora, 1992*). The easily accessible nectaries of *Euphorbia* L. species (Euphorbiaceae) for instance are very frequently visited by ants and an ant role in pollination has been suggested for some members of the genus (*Araf, Kumar & Hamal, 2010*; *Blancafort & Gómez, 2005*; *Schürch, Pfunder & Roy, 2000*).

In the present study we have analysed the pollination system of the endangered forb *E. seguieriana.* An understanding of the pollination system of rare plant species is crucial for their conservation, in particular in species with a high dependency on outcrossing as pollinator conservation may play a key role in management strategies (*Carvalheiro, Barbosa & Memmot, 2008*). Specifically, we assessed the (1) frequency of ants as flower visitors, (2) pollen load and pollen grain viability, (3) the foraging behaviour of ants and hence the likelihood of outcrossing, (4) female fertility as well as (5) the germination rate of seeds. Our results suggest that although ants appear to play a role as pollinators of *E. seguieriana*, they do not contribute to the plant's sexual reproduction.

## MATERIALS AND METHODS

### Study system

Siberian spurge, *Euphorbia seguieriana* subsp. *seguieriana* Necker (Malpighiales: Euphorbiaceae), is a perennial forb, native to Eurasia. In Germany the species inhabits dry, calcareous or sandy grasslands, mainly in the basins of the rivers Rhine, Main, Ems, Nahe, Mosel and Unstrut (*Senghas & Seybold, 2003*). The plant is considered a threatened species and listed in the Red List of the German Federal Agency for Nature Conservation. Siberian spurge typically grows 15–60 cm tall with numerous stems branched from its base. It flowers from June until August and is protogynous, in which the stigma is receptive before the pollen is shed. The inflorescence is a typical cyathium, consisting of strongly reduced male and female flowers, bearing sickle shaped nectaries that offer easily accessible nectar. Functionally the cyathium can be considered as hermaphroditic flower and for simplicity will be termed "flower" here. The capsule fruit is on average 2.3 mm long and bears three seeds which are shed when the capsule dehisces (retrieved from http://www.floraweb.de, 2017).

Two ant species were observed to visit the nectaries of *E. seguieriana*. *Formica cunicularia* Latreille (Hymenoptera: Formicidae; Fig. 1) is a moderately thermophilic species and common throughout Europe where it prefers semidry habitats with a developed herb layer (*Seifert & Schultz, 2009*). Workers are 4.0–6.5 mm long and forage for nectar and honeydew; the species is also zoophagous (*Collingwood, 1979*; *Novgorodova, 2015*). *Tapinoma erraticum* Latreille (Hymenoptera: Formicidae) is a small black ant (2.6–4.2 mm) which lives in diverse habitats, including xerothermic grassland. The species is present in most of Europe; it has a carnivorous and aphidicolous lifestyle and feeds on nectar (*Collingwood, 1979*). The clypeal cleft is considered a morphological adaptation for exploiting liquids hidden in narrow spaces such as floral nectars (*Seifert, 2016*).

The field site was a dry calcareous grassland (480 m$^2$) characteristic for the region of Lower Franconia. The site was situated within the boundaries of the botanical garden of the University of Würzburg, Germany (49°45′54.34″N, 9°55′48.28″E, 214 m above sea level), located on a slope of the Main river valley. The study population of *E. seguieriana* consisted of ca. 300 individuals and was chosen for its ease of access and to limit any negative impact by the experimenter on protected areas. Further populations of *E. seguieriana* exist on calcareous grassland sites along the Main river, which are integral nature reserves (e.g., Grainberg-Kalbenstein, Höhfeldplatte and Scharlachberg) of international significance with a high diversity of endemic and endangered species. Experiments were carried out from 2006–2008.

### Frequency of ants as flower visitors

The frequency of ants and other insects visiting *E. seguieriana* flowers was assessed during a period of 10 observation days in August 2007. Each observation unit consisted of a bundle of rays with approximately 125 cyathia. Twenty of these floral units were observed twice daily (10:00–11:00 and 15:00–16:00) for 1 min per unit and all insects drinking nectar or carrying pollen were recorded. Observations were made at a distance of 80 cm from the inflorescences to avoid disturbance. Specimens of flower visitors were caught with a net

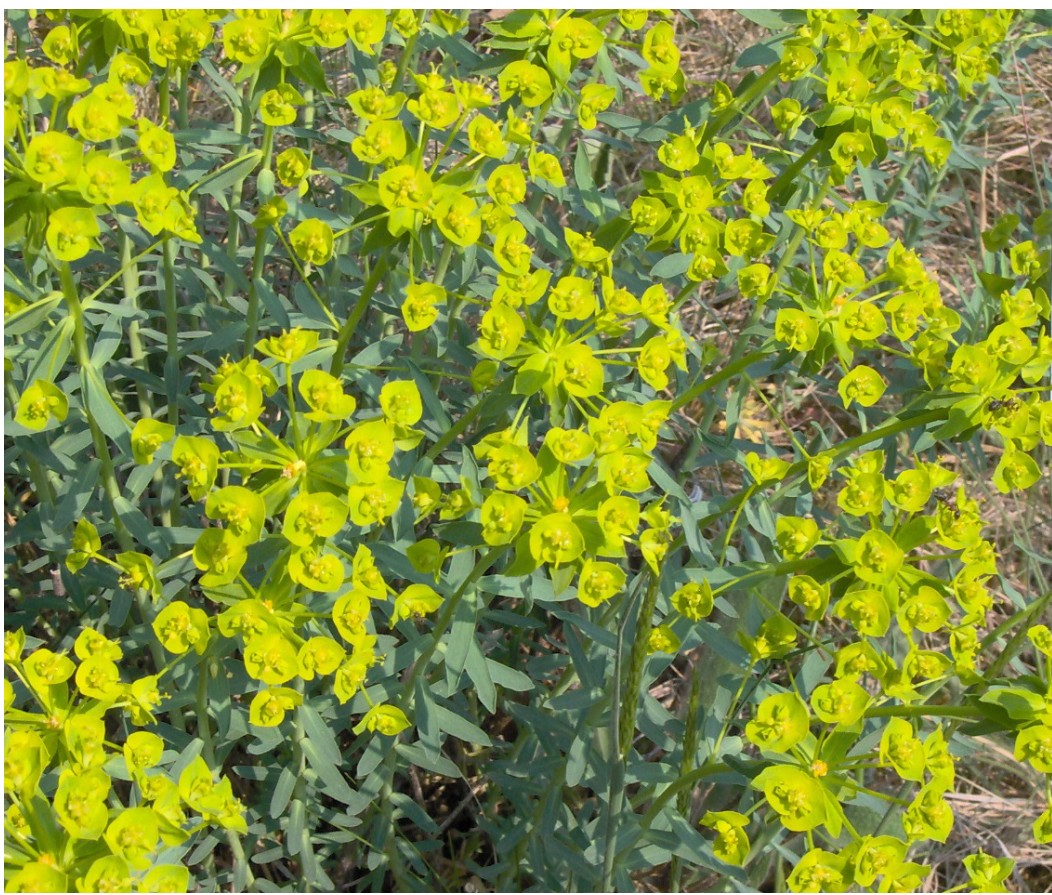

**Figure 1** *Euphorbia seguieriana.* Siberian spurge, *Euphorbia seguieriana* subsp. *seguieriana,* is a perennial forb native to Eurasia. Plants are 15–60 cm tall with numerous stems. The inflorescence consists of strongly reduced male and female flowers (cyathium).

after the observation period. Ants were determined to species level (*Seifert, 1984*; *Seifert, 1996*) while flying insects were identified to the level of family (*Schaefer, 2000*). Voucher specimens of flower visiting ant species were conserved in 70% ethanol and stored in the collection of the Department of Botany, University of Würzburg. The weather during each observation was categorized as sunny, cloudy or rainy.

## Foraging behaviour of ants

The behaviour of individual *T. erraticum* and *F. cunicularia* ants on *E. seguieriana* flowers was compared to obtain more information about their role as potential pollinators. Two to three neighbouring stems of a single plant were marked with a small piece of tape to form one observation unit. Each observation unit consisted of 100 cyathia. Depending on plant size, one or two observation units per plant were designated. Individual ants were observed on 20 units distributed over 13 randomly selected plants for a period of 5 min and at a distance of 80 cm to avoid disturbance. The software The Observer 5.0.25 (Noldus Information Technology, Leesburg, VA, USA) was used to record numbers of

visited cyathia, time spent consuming nectar and frequency of moving between stems of the same plant.

In a second experiment, the potential for outcrossing was assessed by recording whether individual ants exclusively foraged on flowers of the same individual *E. seguieriana* plant or whether other *E. seguieriana* in the vicinity were also visited. Six plants were randomly selected and all foraging ants were carefully marked (paint marker, Edding 750) with a colour dot on the thorax while walking on the flower ($n = 290$ *T. erraticum*, $n = 120$ *F. cunicularia*). A different colour was used for each of the six plants. The following five days, all *E. seguieriana* plants at the field site (approx. 300) were checked daily for the presence of marked ants during 10:00–14:00. The presence of colour-coded ants on the six experimental spurge plants and on other, unmarked *E. seguieriana* was recorded.

## Effect of ants on pollen quality

The quality of pollen attached to the cuticles of foraging ants was assessed using the fluorochromatic procedure (FCR) according to *Heslop-Harrison, Heslop-Harrison & Shivanna (1984)*. Ants foraging on *E. seguieriana* and carrying pollen were allowed to walk onto a petri dish with polytetrafluoroethylene (Fluon; Asahi Glass Co., Ltd., Tokyo, Japan) treated sidewalls to prevent escape and a microscope slide with a drop of FCR solution placed in the centre of the dish. The FCR solution was freshly prepared each day by dropwise adding fluorescein diacetate in acetone ($2$ mg ml$^{-1}$) to 2 ml of a 0.6 M sucrose solution until the solution became slightly cloudy. The trapped ant was held by the legs with insect handling tweezers, placed immediately into the solution with her dorsal side to dislodge all pollen, and was then released. A new microscope slide was used for each ant. Four to five slides were prepared at a time and immediately taken to the laboratory for testing of fluorescent and non-fluorescent pollen numbers using a fluorescence microscope (Leica DMR). The pollen load of 30 ants per species was assessed. As a control, pollen was collected directly from *E. seguieriana* anthers and tested as described above ($n = 23$ plants).

## Effect of ants on plant reproductive success

An exclusion experiment similar to *Schürch, Pfunder & Roy (2000)* was carried out to assess the role of ants as pollinators. Plants in the bud stage were randomly selected and each plant was assigned to one of the following treatments ($n = 10$): (i) All visitors—plants were not manipulated, therefore all insects could visit flowers, (ii) No visitors—Flowers were enclosed by tying mosquito netting fabric around the stems, (iii) Winged only—flying insects could visit the flowers but crawling insects were excluded by driving a plastic cylinder (diameter: 24 cm, height: 24 cm) into the ground that surrounded the plant. The outer cylinder wall was painted with an insect trapping adhesive (Raupenleim grün; Schacht GmbH and Co. KG, Braunschweig), (iv) Ants only—flying insects were excluded by enclosing plants with cages made of mosquito netting fabric and bamboo stick frames. The cage was attached around a cylinder as in (iii) but a 5 cm gap between ground and cylinder rim was maintained to allow ants to access the flowers. Experimental plants were checked daily to ensure that the correct insect groups were excluded. In rare cases where non-target species were found, these were removed immediately. A subsample of five stems

per plant (randomly selected and marked before the experiment) was assessed by counting the numbers of formed fruit capsules and unfertilized female flowers.

The following year the experiment was repeated with the treatments 'Winged only' and 'Ants only' ($n = 9$–$10$). Five stems per plant with approximately similar numbers of cyathia were marked. As soon as the capsules started to ripen and dry out, large empty tea filters (Cilia; Melitta Group Bentz KG, Minden) were tied around the stems. Fruit capsules were left to dehisce and the harvested seeds were then stored for four weeks at 10 °C in a sealed paper bag. Thirty seeds from each plant were sown into trays with seed mix and kept in an open greenhouse tunnel to germinate. After 10 weeks the number of established seedlings was counted.

## Statistical analyses

For each of the individual plants, the numbers of visits by each 'species' (two ant species, *F. cunicularia* and *T. erraticum*, both individually and as a total of all ants, and Winged insects) were totalled over the 10 sampling days and over both morning and afternoon samplings. This total number of visits for each plant and species (and ants as a whole) was then square root transformed to convert it from a Poisson-distributed variable to a variable with an approximate Normal distribution. Four differencing variables were calculated for each plant: (a) visits by *F. cunicularia* vs. *T. erraticum*, (b) visits by *F. cunicularia* vs. Winged insects, (c) visits by *T. erraticum* vs. Winged insects, (d) visits by ants (in total) vs. Winged insects. For each of the four variables, a 95% confidence interval (CI) for the true difference in (square root of) number of visits between the two species was calculated. If this 95% CI did not include zero, this meant there was a 5% significant difference between the two species in visitation rate. In such a case, the level of significance of the difference was determined by calculating the 99% CI, 99.9% CI, and so on. Furthermore, for each of the 20 half-days (10 sampling days × morning/afternoon), the number of visits for each half-day and species (and the three species as a whole) was totalled over the 20 plants and square root transformed. These variables were then input into an analysis of covariance (ANCOVA) routine, with treatment terms being sampling day + morning/afternoon (main effects only) and covariate weather (with 0 = rainy, 3 = cloudy and 4 = sunny). The above four differencing variables were also calculated for each half-day and subjected to the same statistical analysis.

The time ants spent collecting nectar, the frequency of switching between stems, the viability of pollen and the germination rate of seeds were analysed by Mann–Whitney *U* tests. A Kruskal-Wallis ANOVA was used to determine differences in viability of pollen recovered from both ant species and the control.

For the first exclusion experiment overall analysis of variance was considered unsuitable due to the observed data distribution. Instead, the percentage of formed fruit capsules was calculated for each experimental plant; then, the 95% confidence interval for the true mean percentage was calculated for each treatment separately. The data were treated separately since the variability varied widely between the four treatments (two treatments were close to 100%, one treatment was close to 0%, while the fourth treatment had relatively high variability).

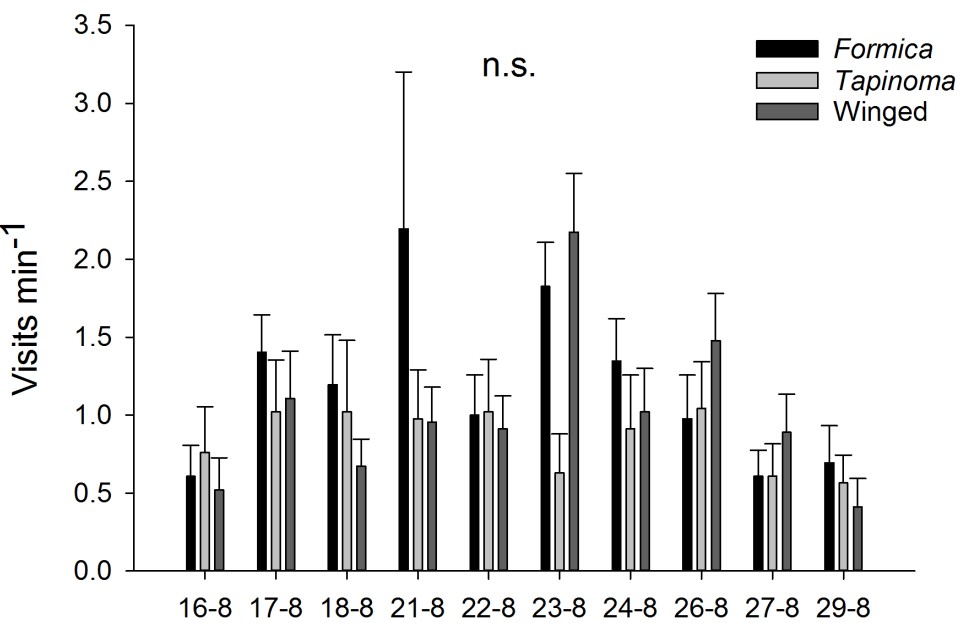

**Figure 2** **Insect visitors of *E. seguieriana*.** Insect visitors of *E. seguieriana* ($n = 20$) during a 10 d observation period. Flowers were censused twice per day for 1 h. Bars show mean ($\pm$s.e.) number of visits per minute. Sampling dates in 2007 are given on the *x*-axis. *Formica*, *F. cunicularia*; *Tapinoma*, *T. erraticum*; Winged, winged insects.

## RESULTS

### Ants as flower visitors

The two ant species that foraged on *E. seguieriana* accounted for 65% (*F. cunicularia*: 36%, *T. erraticum*: 29%) of all flower visitors and were the most frequent visitors ($\sqrt{}$(no. of ants) vs. $\sqrt{}$(no. of winged insects): 95% CI [1.867 $\pm$ 0.595], $p < 0.001$, Fig. 2). The remaining 35% of nectar foraging insects were winged insects of the orders Hymenoptera (Sphecoidea, Halictus) and Diptera (Conopidae, Tachinidae, Syrphidae, Ceratopogonidae). A comparison of flower visitation rates showed no significant differences between the three visitor groups (on the scale of $\sqrt{}$(total no. over 10 days): *F. cunicularia* vs. *T. erraticum*: 95% CI [1.100 $\pm$ 1.399]; *F. cunicularia* vs. winged insects: 95% CI [0.203 $\pm$ 0.643]; *T. erraticum* vs. winged insects: 95% CI [$-0.896 \pm 0.971$]). No significant differences were also found between morning (10:00–11:00) and afternoon (15:00–16:00) visitation rates in the ANCOVAs for any of the eight variables (all species, individual species and pairwise combinations of species including ants vs winged insects, Fig. 3A). However, flower visits correlated positively with weather for the sum of the three species ($p < 0.05$), for *F. cunicularia* ($p < 0.10$) and for winged insects ($p < 0.01$), Fig. 3B. Lower numbers of insects were observed on rainy days and more were observed on sunny days.

### Foraging behaviour of ants

Ants of the larger species *F. cunicularia* visited more than three times as many flowers during each observation period (Mann–Whitney *U* test, $U = 25.5$, $p < 0.001$, Fig. 4A)

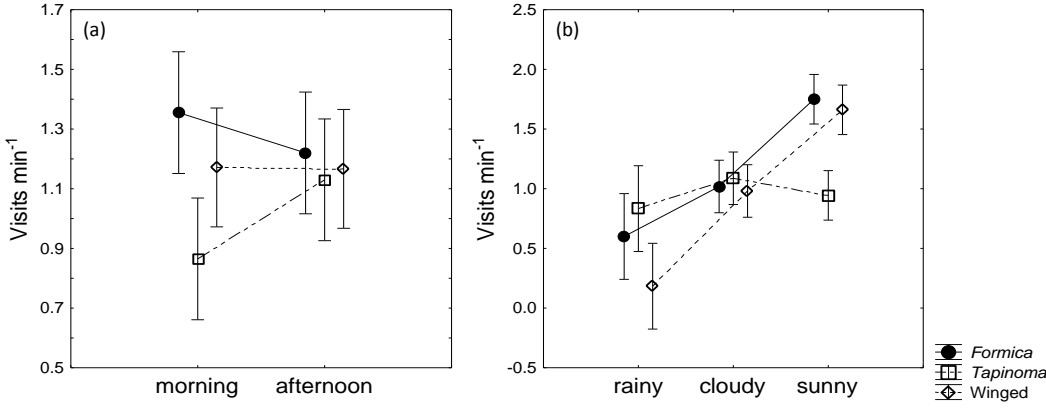

**Figure 3** **Daytime and weather effects on flower visitors.** Influence of (A) daytime and (B) weather on insects visiting cyathia of *E. seguieriana* ($n = 20$) during a 10 day observation period. Symbols depict means and 95% confidence intervals. Formica, *F. cunicularia*; Tapinoma, *T. erraticum*; Winged, winged insects.

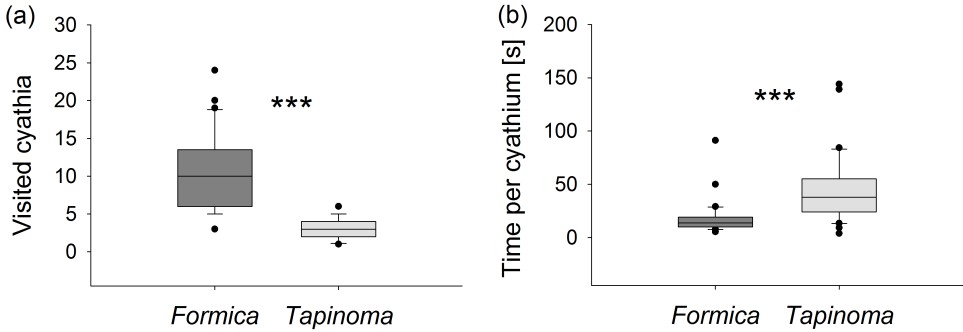

**Figure 4** **Nectar foraging.** (A) Number of visited cyathia and (B) time spent at nectaries by individual ants during a 5 min observation period. Boxes show median (line), 25th and 75th percentiles. Whiskers are 5th and 95th percentiles. Dots show outliers. *Formica*, *F. cunicularia*; *Tapinoma*, *T. erraticum*. Mann–Whitney $U$ test, ***$p < 0.001$; $n = 30$.

but spent only half as much time collecting nectar per flower compared to individuals of *T. erraticum* (Mann–Whitney $U$ test, $U = 154$, $p < 0.001$, Fig. 4B). *F. cunicularia* was also observed to switch more frequently ($1.0 \pm 0.2$ switches per observation) from one stem to another within the same plant than *T. erraticum* ($0.2 \pm 0.1$ switches per observation; Mann–Whitney $U$ test, $U = 245$, $p < 0.001$).

The marking experiment demonstrated that during a five-day observation period both ant species consistently returned to the same individual *E. seguierana* plant on which they had been marked at the start of the experiment. None of the marked ants was found foraging on any other spurge plant in the study population during the daily 4-hour assessments, therefore no statistical tests were carried out. Overall more *F. cunicularia* than *T. erraticum* ants were observed foraging on the plants on which they had been first caught, despite the fact that more *T. erraticum* had been marked initially.

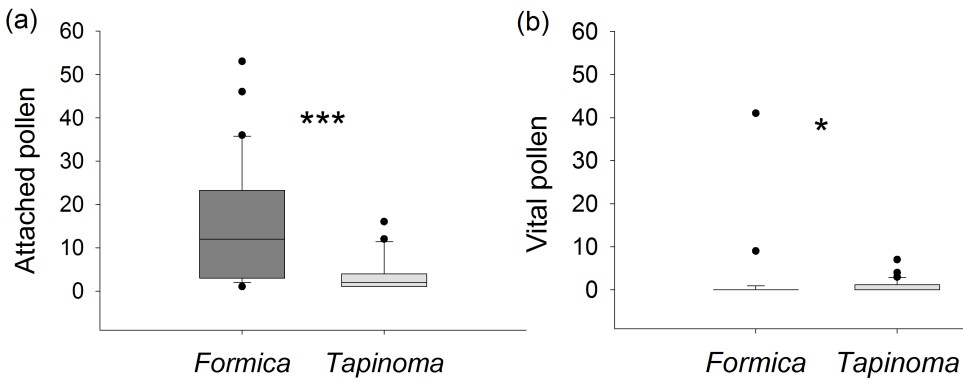

**Figure 5** **Pollen viability.** (A) Number of *E. seguieriana* pollen grains attached to individual ants. (B) Number of attached pollen grains that were viable. Boxes show median (line), 25th and 75th percentiles. Whiskers are 5th and 95th percentiles. Dots show outliers. *Formica* , *F. cunicularia*; *Tapinoma*, *T. erraticum*. Mann–Whitney *U* test, \*$p < 0.05$, \*\*\*$p < 0.001$; $n = 30$.

## Effect of ants on pollen quality

Ants captured on *E. seguieriana* differed significantly in the number of pollen attached to their cuticle (Mann–Whitney *U* test, $U = 147.0$, $p < 0.001$, Fig. 5A). On average, individuals of *F. cunicularia* had pollen loads six times higher than *T. erraticum*. However, in most cases (87%) the pollen sampled from *F. cunicularia* ants did not fluoresce and was considered unviable. The average viability of pollen recovered from all *F. cunicularia* workers was therefore $8 \pm 4\%$. In cases where recovered pollen did show fluorescence (i.e., excluding zero values), the viability rate was $59 \pm 7\%$. In *T. erraticum*, 43% of captured ants carried viable pollen with an average viability rate of $30 \pm 7\%$. In cases where recovered pollen did show fluorescence, the viability rate was $69 \pm 8\%$. A Mann–Whitney *U* test indicated that numbers of viable pollen carried by ants was significantly greater for *T. erraticum* than for *F. cunicularia* ($U = 321$, $p = 0.016$, Fig. 5B). The fluorescence test further showed that pollen recovered from ant cuticles was less viable compared to pollen collected directly from unmanipulated male flowers (viability rate: $72 \pm 4\%$, Table S4) (Kruskal-Wallis test followed by Dunn's tests, d.f. $= 2$, $H = 50.5$, $p < 0.001$, *F. cunicularia* vs plant: $p < 0.001$, *T. erraticum* vs plant: $p < 0.001$).

## Effect of ants on plant reproductive success

Fruit set in *E. seguieriana* depended on the presence and type of insect visitors (Fig. 6). When access to plants was unrestricted for all visitors, nearly 100% of flowers (95% CI [$96.6 \pm 3.5$]) set fruit. Likewise this was found in plants pollinated exclusively by winged insects (95% CI [$98.4 \pm 2.5$]) and therefore no significant difference ($\alpha = 0.05$) in fruit set was observed between treatments 'All visitors' and 'Winged only'. Where access was allowed to ants only, the percentage of flowers turning into seed bearing fruit was $37.2 \pm 18.8\%$ (mean and 95% CI) and therefore significantly lower compared with *E. seguieriana* that received flying visitors. Two 'Ants only' plants failed to produce seeds. When all insects were excluded, only $1.6 \pm 1.5\%$ (mean and 95% CI) of flowers set seed, significantly less than with all other treatments.

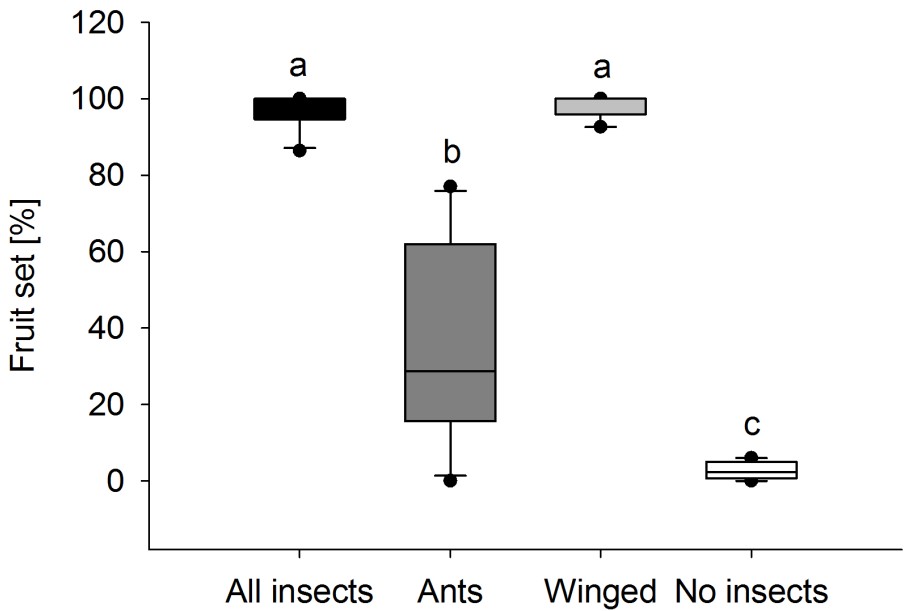

**Figure 6  Fruit set.** Female fertility of *E. seguieriana* measured as percentage of fruit set. All insects, open pollinated plants; Ants, flower access restricted to ants; Winged, flower access restricted to flying insects; No insects, flowers completely enclosed. Boxes show median (line), 25th and 75th percentiles. Whiskers are 5 th and 95th percentiles. Dots show outliers. Different letters indicate significant differences based on the comparison of 95% confidence intervals (*alpha* = 0.05); $n = 10$.

Significant differences were also found in the number of fruit capsules in the second year. 'Winged only' treatment resulted in an average (±SE) of $97.8 \pm 0.2\%$, 'Ants only' in $22.7 \pm 1.7\%$ fruit set per stem (Mann–Whitney *U* test, $U = 0.0$, $p < 0.001$). Seeds from 'Winged only' also had a higher germination rate compared to 'Ants only' plants (Mann–Whitney *U* test, $U = 0.0$, $p < 0.001$, Fig. 7). All of the 'Winged only' plants produced viable offspring with approximately one third of seeds developing into seedlings. In contrast, only one of the 'Ants only' plant produced offspring (germination rate 6.7%). Seeds from all other plants failed to germinate.

## DISCUSSION

This study suggests that ants do not contribute significantly to the reproductive success of *E. seguieriana*. Our findings were somewhat unexpected, because certain aspects suggestive of ant pollination, such as the frequency of ant visits and the proportion of fruit set, had initially indicated otherwise. We found that two ant species were the most frequent visitors of *E. seguieriana* flowers and accounted for two thirds of all recorded insects. A high or very high abundance of ants has been reported as a typical characteristic of ant pollinated plants (e.g., *De Vega et al., 2009*; *Domingos-Melo, Nadia & Machado, 2017*; *Gómez & Zamora, 1992*; *Ibarra-Isassi & Sendoya, 2016*). Workers of *T. erraticum* also showed a fairly constant appearance as their visitation rates were neither influenced by time of day nor weather, quite in contrast to winged insects, which were abundant on sunny days but not when it

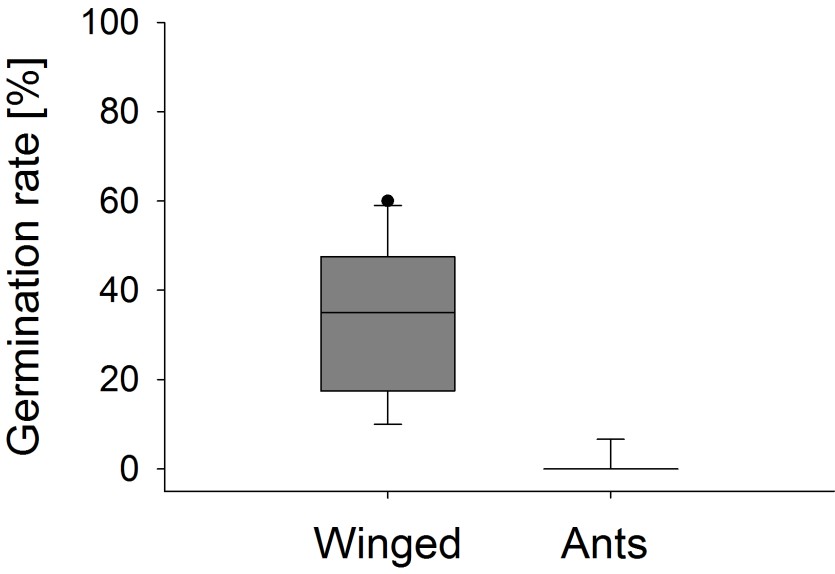

**Figure 7  Seedling emergence.** Germination rate of *E. seguieriana* seeds produced by plants that were pollinated by winged insects and ants, respectively. Boxes show median (line), 25th and 75th percentiles. Whiskers are 5th and 95th percentiles. Dots show outliers. Mann-Whitney *U* test, ***$p < 0.001$, $n = 9$–10.

rained. The fact that ants can move between flowers under adverse climatic conditions has been suggested to promote ant pollination as this makes ants more reliable pollen vectors (*Gómez & Zamora, 1992*; *Hickman, 1974*).

Our observations of nectar-collecting ants indicated that *F. cunicularia* could be a more efficient pollen vector than *T. erraticum*. The former species visited more flowers per observation period and also carried a larger pollen load on its body. However, this notion was contradicted by the fact that only about 10% of all *F. cunicularia* workers, carried pollen grains that were still viable. Contact with the integument of *T. erraticum* also reduced pollen viability but in this species 43% of workers carried viable pollen, thus compensating for the reduced movement between cyathia. Both ants have metapleural glands and it is quite plausible that their secretions were responsible for the adverse effects on pollen viability because viability of pollen from unmanipulated cyathia was significantly higher. Interestingly, several authors reporting successful ant pollination found that pollen viability was not reduced (*Araf, Kumar & Hamal, 2010*; *De Vega et al., 2009*; *Gómez et al., 1996*; *Ibarra-Isassi & Sendoya, 2016*; *Ramsey, 1995*), which suggests that resistance of pollen against metapleural gland secretion could be an adapted trait, enabling plants to use ants as pollinators. Alternatively, pollen transfer in the form of pollinia can be regarded as a preadaptation for successful ant pollination too, as this prevents the pollen grains from contact with the metapleural secretion. Ant pollination has been reported in several species of the pollinia-forming families Orchidaceae and Apocynaceae (*Domingos-Melo, Nadia & Machado, 2017*; *Peakall & Beattie, 1989*; *Schiestl & Glaser, 2012*). In Central Europe two orchid species, *Coeloglossum viride* and *Chamorchis alpine*, are known to be pollinated by ants (*Baumann & Baumann, 2010*; *Claessens & Seifert, 2017*; *Schiestl & Glaser, 2012*).

Individuals of *E. seguieriana* grow as bushy, clump-forming perennials with several stems. Ants were frequently observed to move between stems, which can lead to geitonogamy, if pollen are deposited on the stigmata. Since we were interested in the frequency of ants foraging on surrounding spurge plants and hence on the likelihood of outcrossing, a marking experiment was carried out. The obtained results were unambiguous and showed that over a five day period in which the patch was observed for four hours daily, none of the ant workers were recorded anywhere else than on the individual plant on which they were marked at the beginning of the experiment. This high level of micro-site fidelity has also been reported from other plant-ant associations, such as *Formica pallidefulva* foraging on *Euphorbia esula* (*Fowler, 1983*) or *Camponotus sericeus* visiting extrafloral nectaries of the tree *Pseudocedrela kotschyi* (*Mody & Linsenmair, 2003*), and supports the notion that the occurrence of outcrossing vectored by ants was low in *E. seguieriana*.

As preliminarily indicated by *Rostás & Tautz (2011)*, excluding ants and/or other groups of pollinating insects from *E. seguieriana* flowers showed that ants do contribute to fruit set. Compared with plants exposed to winged insects only, however, the ratio of ovules turning into fruits was considerably lower and no difference was found between open pollinated plants and the former. These findings suggest that ants are not necessary for full fruit set and corroborate those of *Schürch, Pfunder & Roy (2000)* made on *Euphorbia cyparissias* and *Blancafort & Gómez (2005)* on *Euphorbia characias*, who likewise question the importance of ants for reproduction when co-occurring with other visitors. Lower fruit set from pollen grains carried by worker ants was likely the consequence of pollen limitation resulting from exposure to the ants' metapleural secretion. Moreover, limitation can also arise from poor pollen quality (*Aizen & Vázquez, 2006*) and it is conceivable that those pollen grains found to be viable may have nevertheless suffered from exposure to the ants' metapleural secretion. Finally, our results also showed that *E. seguieriana* relies on pollinators, because hardly any fruit formation was recorded from completely enclosed plants, supporting the role of protogyny as a mechanism to prevent selfing.

More crucial than fruit and seed formation for assessing the true relevance of ants in plant sexual reproduction, however, is the number and performance of offspring (*Gómez, 2000*). In this study we found that ten weeks after sowing, one third of seeds resulting from the 'Winged only' treatment had grown into seedlings, while nearly all seeds obtained from ant pollination had failed to germinate. We hypothesise that pollination by flying insects predominantly led to outcrossing and hence to the production of viable seeds, while ant pollination resulted mostly in geitonogamous, infertile seeds. Inbreeding depression is expressed at various life stages, including seed set, germination, and growth and reproduction of offspring (*Angeloni, Ouborg & Leimu, 2011*). Both outcrossing and selfing species can be affected, but in outcrossing plants inbreeding depression is generally more pronounced in the first two stages (*Husband & Schemske, 1996*). To our knowledge, seed germination in conjunction with ants as pollinators has only been measured in *Lobularia maritima* (Brassicaceae) so far (*Gómez, 2000*). In this species no difference in seedling emergence was found, which could be due to the fact that the plant is self-compatible. Furthermore, frequent visits between individual plants were observed, suggesting a certain level of outcrossing. Seed viability has been assessed, either chemically or morphologically

by establishing embryo presence, as an approximation to fitness in two other Mediterranean species (*Blancafort & Gómez, 2005*; *De Vega et al., 2009*). In both cases no differences were found between ant pollinated plants and controls.

## CONCLUSION

Flower visitation by ants does not seem to benefit *E. seguieriana* in terms of reproduction and from the plant's perspective may be characterised as neutral, since no cost in fruit set was measured. The plant has not evolved traits to discourage ants, such as repellent floral volatiles (*Junker, Gershenzon & Unsicker, 2011*) or inaccessible nectaries, which implies the lack of selection pressure. Nevertheless the interaction could be antagonistic, if undetected costs exist, e.g., should ants reduce the number of offspring by lowering the amount of fertile seeds in open pollinated plants compared with plants visited by flying insects only. This aspect needs further evaluation. Our study reiterates the need for assessing the effect of ant pollinators on fitness parameters beyond seed set to accurately estimate the role of ants in plant reproduction, in particular in outcrossing species.

## ACKNOWLEDGEMENTS

We thank Dr Gerd Vogg and members of the Botanic Garden for their support. We also thank Michelle Boyle for critically reading the manuscript.

### Funding

Funding was provided by the Deutsche Forschungsgemeinschaft (DFG), Collaborative Research Centre 554 'Mechanisms and Evolution of Arthropod Behaviour' (project B10) and the Bio-Protection Research Centre at Lincoln University. The funders had no role in study design, data collection and analysis, decision to publish, or preparation of the manuscript.

### Grant Disclosures

The following grant information was disclosed by the authors:
Deutsche Forschungsgemeinschaft (DFG).
Collaborative Research Centre 554 'Mechanisms and Evolution of Arthropod Behaviour'.
Lincoln University.

### Competing Interests

The authors declare there are no competing interests. Dave Saville is founder and employee of Saville Statistical Consulting Ltd. He is contracted by the Bio-Protection Research Centre to support its members in statistical matters.

### Author Contributions

- Michael Rostás conceived and designed the experiments, performed the experiments, analyzed the data, prepared figures and/or tables, authored or reviewed drafts of the paper.

- Felix Bollmann performed the experiments, analyzed the data, authored or reviewed drafts of the paper.
- David Saville analyzed the data, authored or reviewed drafts of the paper.
- Michael Riedel analyzed the data, contributed reagents/materials/analysis tools, authored or reviewed drafts of the paper.

### Supplemental Information

Supplemental information for this article can be found online at http://dx.doi.org/10.7717/peerj.4369#supplemental-information.

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
