# Peer review of "Ants contribute to pollination but not to reproduction in a rare calcareous grassland forb"

_PeerJ, doi:10.7717/peerj.4369_

## Round 0.1 · original submission · Minor Revisions

Both reviewers found this experiments to be well designed and the manuscript to be well written. The reviewers have made minor suggestions for revisions. I agree that figure 5 could be eliminated and replaced by a simple verbal description of the findings in the Results section. Regarding the abstract, I suggest using the term "endangered" or similar on line 12 when the plant is first mentioned. as this can help attract the attention of the conservation community. Actually I'm not sure whether "endangered", "threatened" "Red listed" or "rare" is the best term -- all are used at one time or another in the manuscript. The first sentence of the conclusion (line 312) should start with a capital letter.

·

Basic reporting

.

Experimental design

.

Validity of the findings

.

Additional comments

12 December 2017: Referee's report for PeerJ on
Michael Rostás, Felix Bollmann, David Saville, Michael Riedel
"Ants contribute to pollination but not to reproduction in a rare calcareous grassland forb "
This an interesting and well-conducted study following a clear logic. It confirms in the example of an Euphorbia species that ants are no effective pollinators of plants having no special adaptations to ant pollination. This message is not new but still valuable. However, and this is one of the merits of the paper, the authors also observed seed germination and showed that ants have a clearly deleterious effect on that forb species. The authors also made credible that the high micro-site fidelity of Formica cunicularia should lead to geitonogamy which adds to the well-known deleterious effect of pollen sterility caused by metapleural gland secretions.
Demonstration of pronounced micro-site fidelity of Formica cunicularia was surprising for a myrmecologist. These ants, as many other species of the subgenus Serviformica, do not establish territories, are subordinate to most other, even smaller, ants when encountering them at a food source and were believed to explore their home range in search for food opportunistically and at random. At least the assignation "random" is has to be deleted now – the activity of individual foragers is apparently directed to rather narrow sectors just as in the foragers of the dominant Formica rufa group species (subgenus Formica s.str.) which exploit and defend rich and very long-lasting food sources.
I recommend acceptance of the paper with minor changes or additions. I suggest the following
(a) The authors have stated " We conclude...that ants, although the most frequent visitors, play a negligible role in the reproduction of E. seguieriana." Integrating all facts we know, the authors should better say that ants definitely play a deleterious role in reproduction of Euphorbia segueriana the more as one other point was not addressed by the authors: If one considers that Serviformica ants occupy the flowers for a considerable time and that they are predatory on diverse insects, they will reduce the visiting frequency of the flowers by the true (flying) pollinators – even if it is probably a rare exception that the F. cunicularia can successfully catch such a pollinator on the very top of the plant. The pollinators are chased off at least (see also discussion of this issue in Claessens & Seifert 2017).
(b) There is a rather weak literature survey for sources referring to the area of observation: the Holarctic and Central Europe in particular. The only plants in Central Europe that are specially adapted to ant pollination are the two orchid species Coeloglossum viride and Chamorchis alpina (Baumann & Baumann 2010, Claessens & Kleynen 2011, 2016; Schiestl & Glaser 2012, Claessens & Seifert 2017). As this strongly contrasts the situation described here, the issue should be commented by more than a single sentence.
(c) The message of Fig. 5 is poor. Delete?
Keeping the anonymity of the referee is not necessary.
With best wishes
Bernhard Seifert
Suggested literature:
Baumann, B. & Baumann, H. (2010): Pollination of Chamorchis alpina (L.) Rich. in the Alps by
worker ants of Formica lemani Bondroit: first record of ant pollination in Europe. – J. Eur.
Orch. 42 (1): 3-20.
Claessens, J. & Kleynen, J. (2011): The flower of the European orchid - Form and function. – :
Jean Claessens & Jacques Kleynen, Geulle: 440 pp.
Claessens, J. & Kleynen, J. (2016): Orchidées d'Europe, fleur et pollinisation [Orchids of
Europe, flower and polination]. – Biotope Éditions, Mèze: 448 pp.
Claessens J, Seifert B 2017: Significant ant pollination in two orchid species in the Alps as
adaptation to the climate of the alpine zone? – Tuexenia 37: 363–374.
Schiestl, F.P. & Glaser, F. (2012): Specific ant-pollination in an alpine orchid and the role of
floral scent in attracting pollinating ants. – Alpine botany 122 (1): 1-9.

·

Basic reporting

excellent

Experimental design

creative

Validity of the findings

very convincing

Additional comments

This study examines a rare Euphorbiaceae plant species with flowers that are most frequently visited by ants. The investigators conduct some creative experiments, marking individual ants with different colors on the plants on which they were observed, and saw on subsequent days that the ants never seemed to visit other plants, indicating that all their movements were likely within individual plants (and presumably back to their nests). From this they reasonably conclude that the ants may transfer pollen but only for geitonogamous pollination. Exclusion experiments were employed to demonstrate that plants need visitors to their flowers for fruit set, but that flying insects provide most of the effective pollination (not significantly different from open controls), whereas plants with only ants visiting their flowers set less than one-third of the fruit of other plants. Furthermore, the seeds from those ant-only plants did not germinate well, suggesting there is some inbreeding depression operating.
There were two ants species observed visiting flowers, and one carried much more pollen than did the other. However, a larger proportion of the pollen grains taken from the bodies of the first species were inviable. The scientists implied that the pollen grains were inviable due to the antibiotic secretions of the metapleural glands of the ants, but I wonder: How long is pollen viable when not in contact with ants? And what proportion of fresh pollen is viable? Maybe they know, and did not include this info in the paper. If not, I suggest that it may also be a function of the amount of time spent on the body of the individual ant, for in many plants pollen is viable only for one or two days, and it is possible that pollen stays on an ant so long it may simply be old.
This elegant study was quite comprehensive, but only the pollen viability question was left it my mind. Perhaps the authors can comment on that possibility or provide some more evidence that it was the ant secretions that rendered the pollen inviable. Actually I see from the data table on pollen viability, they evidently present data on the natural proportions of viable pollen from flowers – is that correct? And so just a little more comparison and explanation would be helpful.
In Fig. 1 caption - I think the term ‘cyathium’ should be used as it is one only that has the ant on it.
It may be reasonable to eliminate Figure 5.
Fig. 8 caption – if * means that, what do *** mean? Maybe better to put different letters above the bars.

---

## Round 0.2 · Minor Revisions

The reviewers' comments have been addressed and I think this is a nice piece of work. I have some minor comments / edits to consider as follows:

Line 27 “ viability of pollen on ant cuticles was very low” -- I suggest replacing the subjective term “very low” with actual numbers or use a more objective descriptor like “significantly lower (P < 0.001)”.
The reported viability rate for fresh pollen was 72% versus 69% or 58% from ants; (but these figures contrast with figure 5b which suggests a lower mean. Maybe the lower mean (from all ants) should also be presented in the written Results?

Line 31 for clarity, “almost none,” to “almost none set fruit” (removing comma after none)

Line 36 Just a suggestion to consider rewording the concluding phrase, which doesn’t sound quite right to me: “investigate postdispersal plant stages in order to confirm ant-plant mutualisms.” Do you mean something like “investigate plant fitness effects beyond seed set in order to confirm ant-plant pollination mutualisms”? Alternatively, the wording used at the end of the conclusion section (line 350) seemed clear to me.

Line 58 “ Ortstreue” – I guess this is capitalized because all nouns in the German language are capitalized?

Line 73 “a role” to “an ant role”

Line 82 “ in E. seguieriana” to “ of E. seguieriana”

Line 143 “ every E. seguieriana plant” to “ all E. seguieriana plants” (to match the verb “were”)

Line 265 “significantly lower” give P-value and test statistic

Line 267 Same comment as above.

Line 284 “ daytime” do you mean “time of day”?

---

## Round 0.3 · accepted · Accept

Minor revisions were carefully completed.